# Double superionicity in icy compounds at planetary interior conditions

Kyla de Villa [1] ✉, Felipe González-Cataldo [1] & Burkhard Militzer [1,2]

The elements hydrogen, carbon, nitrogen and oxygen are assumed to comprise the bulk of the interiors of the ice giant planets Uranus, Neptune, and sub-Neptune exoplanets. The details of their interior structures have remained largely unknown because it is not understood how the compounds $H_2O$, $NH_3$ and $CH_4$ behave and react once they have been accreted and exposed to high pressures and temperatures. Here we study thirteen H-C-N-O compounds with ab initio computer simulations and demonstrate that they assume a superionic state at elevated temperatures, in which the hydrogen ions diffuse through a stable sublattice that is provided by the larger nuclei. At yet higher temperatures, four of the thirteen compounds undergo a second transition to a novel doubly superionic state, in which the smallest of the heavy nuclei diffuse simultaneously with hydrogen ions through the remaining sublattice. Since this transition and the melting transition at yet higher temperatures are both of first order, this may introduce additional layers in the mantle of ice giant planets and alter their convective patterns.

H-C-N-O chemistry is expected to dominate the interior structures of ice giant planets, such as Uranus, Neptune, and perhaps many sub-Neptunes, one of the most common planetary types in our galaxy[1]. While there is still considerable ambiguity in the exact structure and composition of ice giant planets[2], the formation of these planets beyond the frost line has been the basis of the assumption that these planets have a solar composition of H:O:C:N of ~28:7:4:1; this composition, which has been studied experimentally and computationally, is referred to as synthetic Uranus[3,4]. Interior models of ice giants suggest that they consist of small rocky cores, H-He rich atmospheres, and thick mantles of hot planetary ices, formed at high pressures from accreted $H_2O$, $CH_4$, and $NH_3$[5].

At the high pressures (~10–700 GPa) and temperatures (~2000–6000 K) of Uranus and Neptune[6], water and ammonia exhibit superionic states, in which hydrogen ions diffuse like a liquid through stable lattices of heavier nuclei[7]. The high ionic conductivity of these phases, in combination with electronic conductivity, may be significant for the generation of the unusual non-dipolar, non-axisymmetric magnetic fields of Uranus and Neptune[8,9].

Superionic materials have garnered much interest in recent years because of their potential to contribute to planetary dynamos,

with studies on superionic iron alloys[10–16]; helium-methane[17], helium-ammonia[18], and helium-silica compounds[19]; magnesium oxides[20] and magnesium hydrosilicates[21]; silica-water and silica-hydrogen compounds[22]; and hydrogen-oxygen compounds[23]. Studies of superionic phases of planetary ices have primarily focused on water, ammonia, and ammonia-hydrates[6,7,24–41], and have not identified hydrogen superionicity in carbon-bearing ices despite expectations that carbon is also a major constituent of ice giant interiors. The chemical space of planetary ices is likely much broader than studied so far.

Computational chemical structure searches have led to the discovery and subsequent synthesis of many compounds, such as novel high $T_c$ superconductors $LaH_{10}$ and $H_3S$[42]. For an understanding of possible chemical structures stable at ice giant interior conditions, we turn to structure searches of the H-C-N-O quaternary space. Conway et al.[43] and Naumova et al.[44] predicted a number of stable high-pressure phases across the H-C-N-O system, based on enthalpy calculations from density functional theory simulations performed at 0 K. Naumova et al. additionally performed phonon calculations to determine the dynamic stability of their predicted compounds up to 3000 K.

[1]Department of Earth and Planetary Science, University of California, Berkeley, CA 94720, USA. [2]Department of Astronomy, University of California, Berkeley, CA 94720, USA. ✉e-mail: kyla.devilla@berkeley.edu

We performed density functional molecular dynamics (DFT-MD) simulations to explore the effects of temperature on the thirteen H-bearing H-C-N-O ices predicted by these works[43,44]. In addition, we generated machine learning potentials for a subset of these ices, which enabled us to perform simulations with far greater timescales and lengthscales than would be computationally feasible using DFT-MD.

With these two simulation methods, we find that four out of thirteen of the H-C-N-O ices we studied exhibit *double superionicity*: $H_3NO_4$-$C2_1$ (400 GPa)[44], $H_3NO_4$-$P2_12_12_1$ (450–600 GPa), HCNO-$Pca2_1$-II (500 GPa), and $CH_2N_2$-$Pna2_1$ (500 GPa)[43]. As temperature increases, these systems transition from a solid to a hydrogen superionic state where H ions diffuse through a stable lattice of the remaining elements. Then, at higher temperatures, a second species (either C or N) becomes diffusive while the remaining, heavier atomic species maintain a stable sublattice. While C and O have been shown to exhibit superionic diffusion in Fe alloys[10], and previous works have demonstrated multi-element superionicity in other materials (H and He for high-pressure He-$H_2O$ systems[45], and Ag/Cu and Hg in superionic conductors $Ag_2HgI_4$/$Cu_2HgI_4$[46]), neither heavy-element nor multi-element superionicity has previously been observed in H-C-N-O systems.

## Results

### Hydrogen superionicity in planetary ices

Using DFT-MD simulations, we heated thirteen H-bearing H-C-N-O ices (see Fig. 1) with pressures of 50–500 GPa at 0 K[43,44], in 500 K increments at constant density. For each temperature, we calculated the

mean squared displacement (MSD) for each atomic species, defined as

$$MSD(t) \equiv \langle (\mathbf{r}_i(t) - \mathbf{r}_i(0))^2 \rangle = 6Dt, \tag{1}$$

where $\mathbf{r}_i(t)$ is the position of ion $i$ at a time $t$, $\mathbf{r}_i(0)$ is its initial position, and the MSD at each time is averaged over all ions of a given atomic species. $D$ is the temperature-dependent self-diffusion coefficient for a given species. We expect $D \approx 0$ for non-diffusing species, and $D > 0$ for diffusing species.

In all H-bearing ices we studied, we observed a solid to hydrogen superionic phase transition with increased temperature, characterized by $D_H > 0$ while $D \approx 0$ for all other atomic species. The majority of these ices (ten out of thirteen) begin to exhibit hydrogen superionic behavior at 1500 to 2000 K, consistent with the behavior of $H_2O$ at extreme pressures[27], but we also observed solid to hydrogen superionic phase transitions at temperatures as low as 1000 K and as high as 3500 K. In Fig. 1a we show the relationship between the solid to hydrogen superionic transition temperature ($T_s$) and the fraction of hydrogen atoms in each of the thirteen H-C-N-O ices. Hydrogen rich ices exhibit H diffusion at lower temperatures (1000–2000 K), whereas hydrogen poor ices must reach higher temperatures (2000–3500 K) before H ions begin to diffuse. These hydrogen poor ices (red, black) have a greater number of heavy ions (C, N, and O) around H ions at short distances compared to hydrogen rich ices (yellow, green), as shown in Fig. 1b. As proton fraction decreases, H ions become increasingly constrained, and they require higher temperatures to be reached before they can acquire the kinetic energy needed to escape their potential wells.

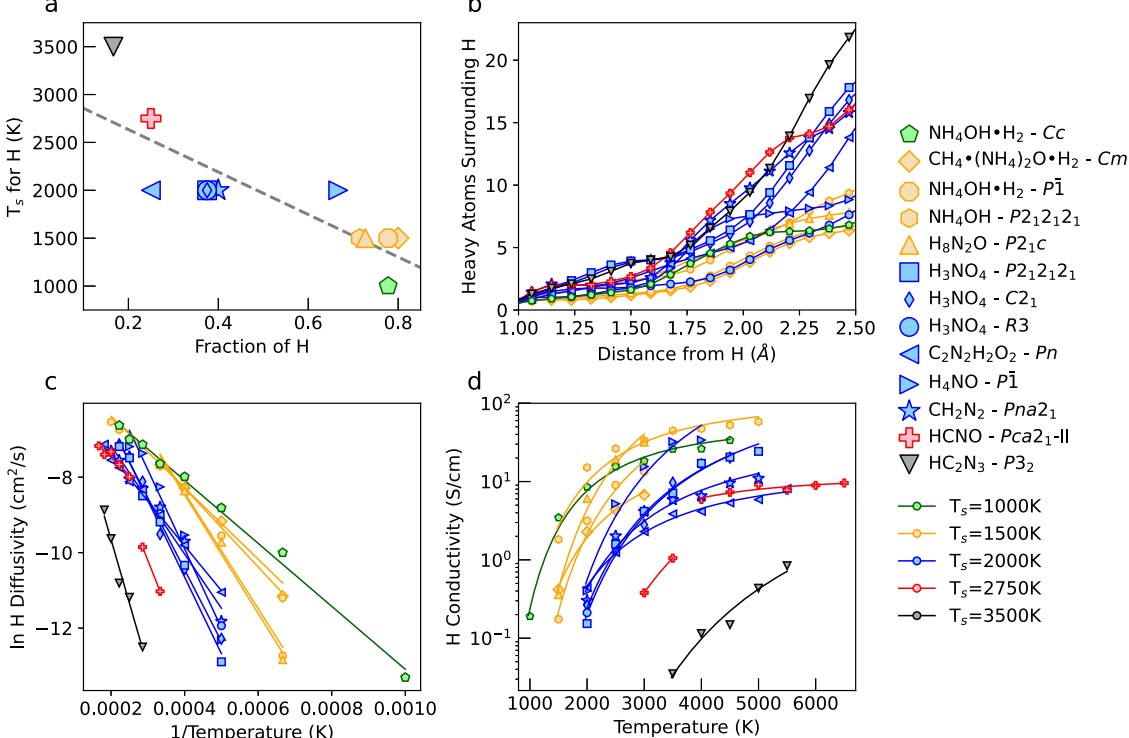

**Fig. 1 | Effects of proton fraction on hydrogen superionic transition temperature, diffusivity, and ionic conductivity for H-C-N-O ices. a** Markers show the lowest temperature at which hydrogen superionicity is observed for H-C-N-O ices ($T_s$) as a function of their fraction of hydrogen ions ($N_H/N$). The dashed gray line gives a linear fit for all points which emphasizes the overall trend that materials with a high hydrogen content require lower temperatures to mobilize H ions into a superionic state. For (**b**–**d**), line colors correlate to $T_s$ as shown in (**a**). **b** For each material we show the average number of C, N, and O ions within a given distance of H ions. Materials with fewer heavy ions around H ions exhibit lower transition temperatures to hydrogen superionic states. **c** Diffusivity of H ions for each material plotted as Arrhenius trends with $\ln(D_H)$ vs. $1/T$. Materials in which H ions first diffuse at higher temperatures have steeper slopes, indicating a higher activation energy. **d** Ionic conductivities for H for superionic phases. Materials with lower temperature superionic phases have higher conductivities. HCNO-$Pca2_1$-II (shown in red) is best represented using two linear functions, with the lower temperature hydrogen superionic phase, and the higher temperature doubly superionic phase considered separately.

In Fig. 1c we show the H ion diffusivity of the H-C-N-O materials within the hydrogen superionic and doubly superionic phases. We find H superionic diffusion in all materials to follow an Arrhenius trend[47]:

$$\ln(D_H) \propto -E_a/T \tag{2}$$

where $E_a$ is the activation energy required for an H ion jump. Materials with a higher $T_s$ (red, black) have greater slopes when plotted as $\ln(D_H)$ vs. $1/T$ compared to those with a lower $T_s$ (yellow, green), implying that a higher activation energy is required for H ions to start diffusing. Moreover, materials with a lower $T_s$ show higher diffusivities at any temperature, indicating there are a greater number of diffusion pathways within the cell for the H ions to travel through.

Diffusivities for HCNO-$Pca2_1$-II are best represented using two linear functions, with the lower temperature hydrogen superionic phase, and the higher temperature doubly superionic phase considered separately. Once the C ions begin to diffuse, the slope of the $\ln(D_H)$ vs. $1/T$ line decreases, indicating that the potential energy landscape controlling the rate of H ion diffusion has changed and that H ions can move more easily within the lattice.

Finally, we calculated the ionic conductivity from H ion diffusion for the hydrogen superionic and doubly superionic phases of all thirteen ices (Fig. 1d) using the Nernst-Einstein relation, which links ionic conductivity $\sigma$ to diffusivity $D_H$ (calculated from MSD), ionic charge $q$ of H ions (calculated using the Bader Charge Analysis—see Fig. S4)[48–51], temperature, Boltzmann's constant, and the concentration $c$ of diffusing H ions in the simulation cell ($N_H$/volume):

$$\sigma = D_H c q^2 / k_b T \tag{3}$$

H-C-N-O materials which become hydrogen superionic at lower temperatures have higher ionic conductivities, which is largely controlled by the higher ionic diffusivities of these materials. We found the thirteen H-C-N-O ices to have ionic conductivities from ~0.01–100 S/cm, consistent with ranges previously calculated for superionic water and ammonia[7,38,52]. The high ionic conductivities of the superionic phases of these H-C-N-O materials suggests that these or similar H-bearing ices could be important for planetary dynamos if they were present in the proposed convecting region of ice giant planets responsible for magnetic field generation.

We observe hydrogen superionicity in a number of N-H-O ices, corroborating previous works which have observed hydrogen superionicity in the chemical space spanned by $H_2O$-$NH_3$ compounds[29,32,34]. However, here we also report hydrogen superionicity in carbon-bearing ices for the first time. Previous studies have indicated that hydrocarbons under high pressure would undergo dehydrogenation[53] and diamond formation[54], preventing the formation of a superionic state. In the ices we studied, the presence of O or N stabilizes the C sublattice, allowing these C-bearing ices to exhibit superionic diffusion of hydrogen ions.

With further heating, nine out of thirteen of these ices transition from the hydrogen superionic state directly to the liquid phase ($D > 0$ for all species), with the entire sublattice of heavy elements melting at once.

## Superionic diffusion of multiple elements

In four of the thirteen H-C-N-O ices we studied, following the transition from solid (no diffusion) to hydrogen superionic ($D_H > 0$), a further transition occurs in which a second ion type starts to diffuse. The remaining one or two heavier species provide a sublattice for the two lighter elements to diffuse through. We monitored these doubly superionic phases in DFT-MD simulations in the NVT ensemble for 40–70 ps and found the diffusive behavior of the two lightest elements and the stability of the lattice of the remaining element(s) to persist on these long timescales.

To confirm the long-term stability of the doubly superionic behavior we observed, we generated machine learning potentials (MLPs) for each of the four doubly superionic ices using the Deep Potential for Molecular Dynamics (DeePMD) method[55,56]. Our MLPs were trained to replicate chemical behavior for the pressure and temperature conditions explored by constant density simulations of each ice when interfaced with LAMMPS[57]. With our MLP-MD simulations, we were able to confirm the existence of the doubly superionic phases we observed in DFT-MD simulations, and replicate the transition temperatures for all phase changes for all four doubly superionic ices. We found the doubly superionic phases of each ice to be stable across many independent trajectories and long ($t \approx 100$ ps) timescales, at all temperatures observed with DFT-MD simulations.

We classify solid phases as having near zero diffusion for all elements; hydrogen superionic phases as exhibiting only H diffusion; doubly superionic phases as exhibiting only diffusion of H and a second, heavier element; and liquid phases as exhibiting diffusion of all elements.

Figure 2a–c shows the diffusion behaviors of each atomic species in $H_3NO_4$-$P2_12_12_1$ as the system is (a) solid, exemplified by the near zero slope of the MSD for all elements (oscillations in the MSDs are due to atomic vibrations within the stable lattice), (b) hydrogen superionic, demonstrated by the positive MSD slope for H diffusion but the near zero slope of the MSD trends for N and O, and (c) doubly superionic, shown by positive MSD slopes for both H and N yet a near zero slope of the MSD of O.

The atomic trajectories of each of these three diffusion regimes are shown for $H_3NO_4$-$P2_12_12_1$ in Fig. 2d–f. We observe that at 500 K, N and O both experience minimal displacements within their potential wells, but that H atoms vibrate significantly about their equilibrium positions. At 3000 K, we observe that N and O experience greater vibrations than at lower temperatures, but are both still confined to their potential wells, while H diffuses throughout the N-O lattice, exhibiting the hydrogen superionic phase. At 4500 K, N ions are also mobile, diffusing throughout the stable fcc-lattice of O atoms, while H ions maintain their diffusive behavior but with much greater diffusivities than at 3000 K.

Figure 2g–i shows isosurfaces of the most probable locations of superionic H ions for 3000 K in the hydrogen superionic phase, and H ions and N ions at 4500 K in the doubly superionic phase, demonstrating the diffusion pathways of superionic H and N through the stable lattice. At 3000 K, H ions diffuse through the faces of $O_4$ tetrahedra. At 4500 K, N ions are also diffusing, and H and N diffuse within the fcc O lattice in interwoven pathways.

While we observe linearly increasing MSD trends for at least one temperature for each doubly superionic material, for several of these materials, the second diffusive element exhibits hampered diffusion before becoming fully diffusive at high $T$. Trajectory plots revealed that at these lower temperatures, although some of the ions of the second superionic species diffuse through the cell, this diffusion is infrequent and often involves atoms hopping back and forth from their original lattice site and a nearby interstitial site, and less frequently, diffusing more broadly through the cell. We still classify materials as doubly superionic if we observe any diffusion of the second superionic species. As we discuss later, we observe pressure and/or enthalpy discontinuities at fixed density for each ice upon any diffusion of the second superionic element, affirming these phase boundaries.

In Fig. 3 we show the diffusion coefficients for each material for both DFT-MD and MLP-MD simulations. With our MLPs we calculated comparable elemental diffusivities and MSD trajectories to those derived from DFT-MD simulations.

By 3000 K, all four doubly superionic ices exhibit hydrogen superionicity with a characteristic ~$10^{-4}$ cm²/s hydrogen diffusivity[7], while the diffusivity of all other species is low (~$10^{-7}$ cm²/s). The doubly

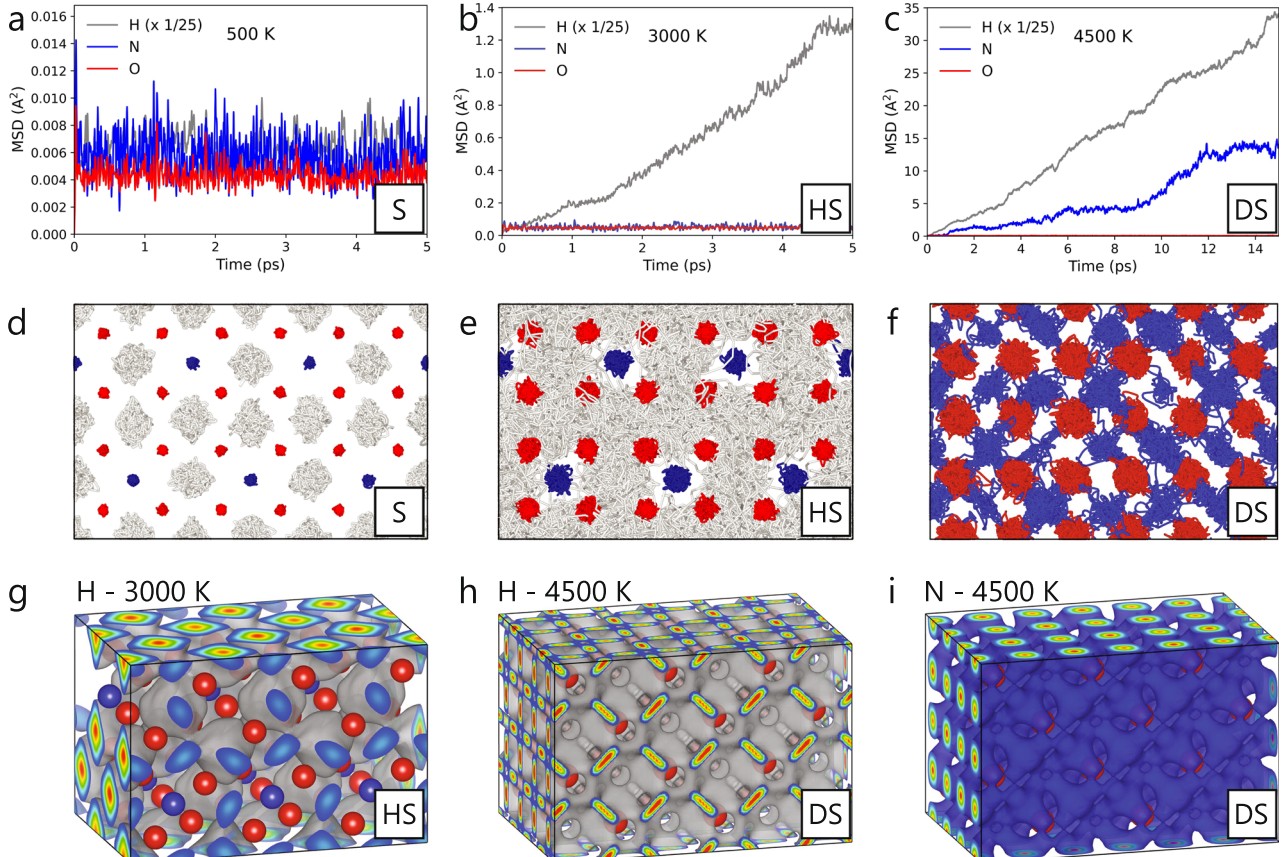

**Fig. 2 | Visualization of H and N diffusion in $H_3NO_4$-$P2_12_12_1$.** Mean squared displacement (MSD) trends for all elements for (**a**) solid (S), (**b**) hydrogen superionic (HS), and (**c**) doubly superionic (DS) phases. For clarity the MSD for H has been multiplied by 1/25 in each plot. **d**–**f** Atomic trajectories over 5, 5, and 10 ps, respectively, for corresponding temperatures (above). In (**f**), H trajectories have been omitted to better visualize N diffusion. **g**-**i** Isosurfaces show the most probable locations of superionic H (gray) at 3000 K and 4500 K, and superionic N (blue) at 4500 K throughout the cell. Results are shown for a 96 atom cell to allow for easier visualization of superionic isosurfaces. All results shown in this figure were derived from density functional molecular dynamics (DFT-MD) simulations. See Figs. S5–7 for panels **a**–**f** for the other three doubly superionic materials.

superionic phase is distinguished by elevated diffusivities of the second diffusive element: N for $H_3NO_4$-$C2_1$ and $H_3NO_4$-$P2_12_12_1$, but C for $CH_2N_2$-$Pna2_1$ and HCNO-$Pca2_1$-II. The liquid phase is distinguished by high diffusivities of all elements.

$H_3NO_4$-$P2_12_12_1$ (500 GPa, 6.033 g/cm³) becomes hydrogen superionic at 2000 K and doubly superionic at 4000 K. This ice maintains N and H superionic diffusion through a stable fcc O lattice through 4500 K. At 5000 K the O lattice melts, transitioning the material to a liquid phase.

In $H_3NO_4$-$C2_1$ (400 GPa, 5.644 g/cm³), similarly to $H_3NO_4$-$P2_12_12_1$, H ions diffuse at 2000 K. However, in this lower pressure $C2_1$ phase, N ions diffuse at a slightly lower temperature, 3500 K. The fcc O lattice of the $C2_1$ phase also melts at 5000 K.

Summaries of phase transitions (**bolded** elements are mobilized) are given below:

$$\text{H}_3\text{NO}_4 \xrightarrow[\text{2000 K}]{\text{H mobilized}} \boldsymbol{H}_3\text{NO}_4 \xrightarrow[\text{3500/4000 K}]{\text{N mobilized}} \boldsymbol{H_3N}\text{O}_4 \xrightarrow[\text{5000 K}]{\text{O mobilized}} \boldsymbol{H_3NO_4}$$
$$\text{solid} \qquad\qquad \text{superionic} \qquad\qquad \text{doubly superionic} \qquad\qquad \text{liquid}$$

$CH_2N_2$-$Pna2_1$ (500 GPa, 5.311 g/cm³) exhibits H superionicity from 2000–4500 K, and C superionicity from 4750–5250 K. The hcp N sublattice melts at 5500 K.

$$\text{CH}_2\text{N}_2 \xrightarrow[\text{2000 K}]{\text{H mobilized}} \boldsymbol{C}\text{H}_2\text{N}_2 \xrightarrow[\text{4750 K}]{\text{C mobilized}} \boldsymbol{CH_2}\text{N}_2 \xrightarrow[\text{5500 K}]{\text{N mobilized}} \boldsymbol{CH_2N_2}$$
$$\text{solid} \qquad\qquad \text{superionic} \qquad\qquad \text{doubly superionic} \qquad\qquad \text{liquid}$$

HCNO-$Pca2_1$-II (500 GPa, 5.977 g/cm³) becomes hydrogen superionic at 2750 K. This ice exhibited the widest temperature range of double superionicity, from 4000–6250 K. The hcp N-O sublattice melts at 6500 K.

$$\text{HCNO} \xrightarrow[\text{2750 K}]{\text{H mobilized}} \boldsymbol{H}\text{CNO} \xrightarrow[\text{4000 K}]{\text{C mobilized}} \boldsymbol{HC}\text{NO} \xrightarrow[\text{6500 K}]{\text{N and O mobilized}} \boldsymbol{HCNO}$$
$$\text{solid} \qquad\qquad \text{superionic} \qquad\qquad \text{doubly superionic} \qquad\qquad \text{liquid}$$

We performed a number of additional simulations of the $H_3NO_4$-$P2_12_12_1$ structure, shown in Fig. 4, to confirm that the doubly superionic phase could persist when simulated with larger system sizes, is stable against supercell deformation, and is energetically stable relative to the liquid phase. Following NVT simulations of a 96 atom cell with a 0 K pressure of 500 GPa, we carried out additional DFT-MD simulations on a larger, 288 atom cell, in the NVT, NPT (Figs. S8–9), and NPH ensembles. These simulations repeatedly and unambiguously replicated the doubly superionic behavior, only indicating a 500 K lower melting temperature than observed with a 96 atom cell.

We also performed two-phase simulations (Figs. S10–11) of $H_3NO_4$-$P2_12_12_1$ with the liquid and doubly superionic phases using DFT-MD in the NVT ensemble. These simulations well replicated the melting point that we obtained with other simulation methods. For 4000 K and 4500 K, the doubly superionic phase grew by crystallizing O ions from the liquid into an fcc sublattice while allowing the H and N ions to continue to diffuse superionically. For 5000 K, the doubly superionic

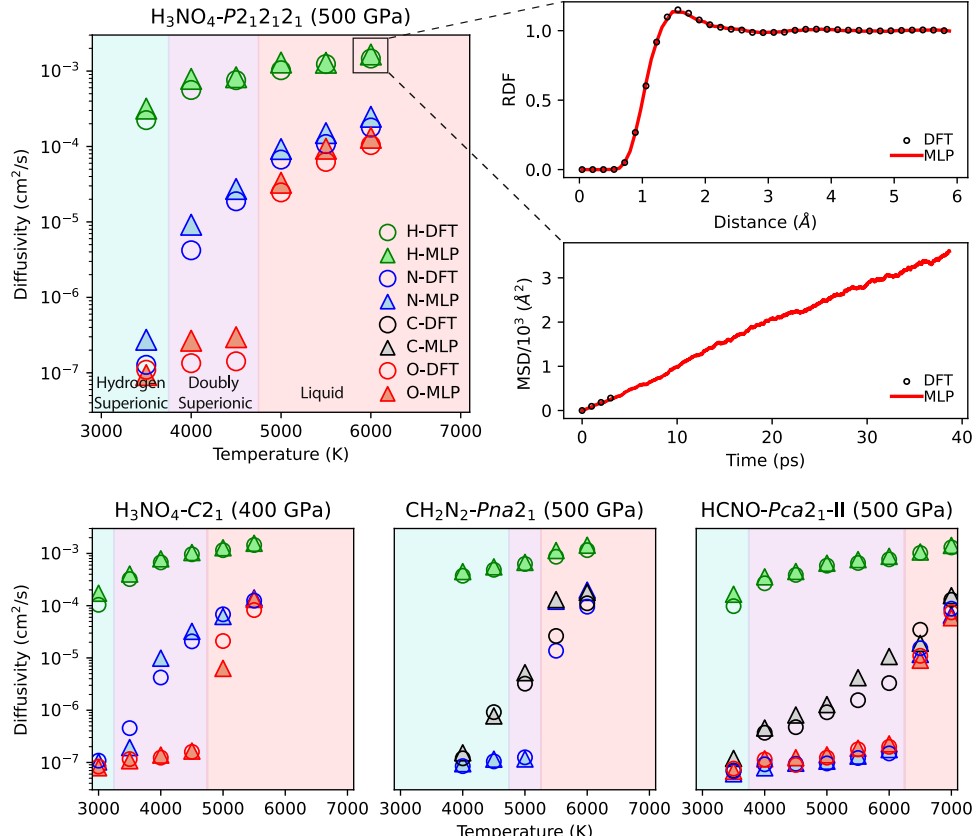

**Fig. 3 | Comparison of elemental self-diffusivities calculated with density functional molecular dynamics (DFT-MD) and machine learning potential molecular dynamics (MLP-MD).** Diffusivities for $H_3NO_4$-$C2_1$ (144 atoms), HCNO-$Pca2_1$-II (128 atoms), and $CH_2N_2$-$Pna2_1$ (160 atoms) were calculated from the average of five independent trajectories for both DFT and MLP. Diffusivities for $H_3NO_4$-$P2_12_12_1$ (288 atoms) were calculated from one trajectory due to time limitations from the larger system size. Shading indicates the phase of the material at a given temperature: cyan is superionic H, purple is superionic H + C or N, and red is liquid. Marker color indicates elements: green is H, black is C, blue is N, and red is O. Circular markers are used for DFT-MD and triangular markers are used for MLP-MD. For $H_3NO_4$-$P2_12_12_1$ at 6000 K, we show a comparison between DFT and MLP of the H MSD trend, and the associated H-H radial distribution functions.

phase melted and the entire cell became liquid. Finally, ML-MD simulations of 1920 atoms and more than 100 ps validated the stability of the doubly superionic phase using a large system size and long time scales. See Movies S1–2 for videos of our two-phase simulations and large cell doubly superionic diffusion.

## Superionic diffusion mechanisms

The classical picture of superionic diffusion in solids involves ionic species jumping through a series of energetically similar potential wells. These wells include the original positions of those ions, as well as vacancies and interstitial sites in the stable lattice[47]. For atomic diffusion to occur by this well-hopping mechanism, there must be more sites than diffusing ions. We find that the diffusion of heavier species in the doubly superionic state occurs via this mechanism. Upon the melting of the H lattice in each of these ices, a number of new sites become energetically accessible for the heavy superionic element if the material is sufficiently heated.

In $CH_2N_2$-$Pna2_1$, hydrogen superionicity is also enabled by H diffusion through interstitial sites. The mechanisms of H diffusion within the $H_3NO_4$ ices and HCNO-$Pca2_1$-II, however, are not as straightforward, and change upon the diffusion of the second superionic element. In the hydrogen superionic phase, there are no observable vacancies nor interstitial sites into which H ions can hop and linger. Vibrating around in their potential wells, H ions will infrequently traverse into a neighboring potential well, and for a short time two H ions will be present in the same potential well. Following this, either the traveling H ion will return to its original site, or the H ion whose well became doubly occupied would itself travel to a new well by the same mechanism. Upon the transition to the doubly superionic phase, interstitial sites become accessible to the H ions, which adopt new diffusion pathways.

## Thermodynamic properties

In Fig. 5, we show the pressure and internal energy dependence on temperature from constant density simulations of each doubly superionic ice. For each phase, there is an approximately linear relationship between pressure and temperature, and internal energy and temperature, whose slopes represent the thermal pressure and specific heat, respectively. Moreover, for all four materials, we see that the pressure-temperature and internal energy-temperature curves for the solid and the hydrogen superionic phases exhibit similar slopes and have little to no offset between them.

For the hydrogen superionic to doubly superionic phase transition, and the doubly superionic to liquid phase transition, however, we observe offsets in the pressure and/or internal energy implying that these transitions are of first order and associated with discontinuities in density and/or latent heat. (The temperature conditions of the discontinuities in pressure and/or internal energy match well with the phase transitions observed by studying diffusion of different atomic species).

$H_3NO_4$-$C2_1$ and $H_3NO_4$-$P2_12_12_1$ both show a slight increase in pressure and internal energy upon the transition to the doubly superionic state, which is followed by much larger offsets in pressure and internal energy upon melting.

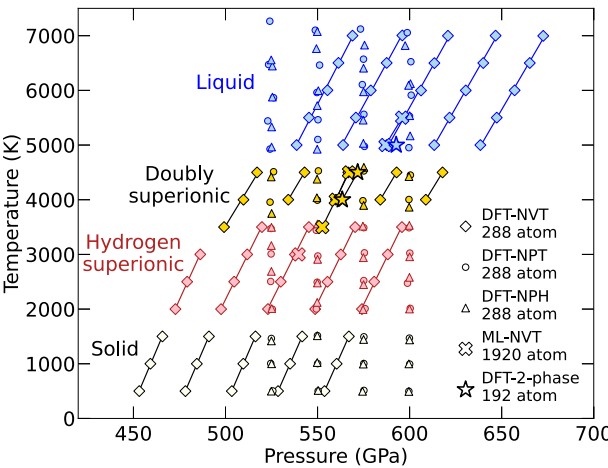

**Fig. 4 | Simulations of doubly superionic H₃NO₄-P2₁2₁2₁.** We performed a number of NVT, NPT, NPH, and 2-phase simulations using density functional molecular dynamics (DFT-MD), in addition to large cell machine learning molecular dynamics (ML-MD) simulations, to confirm the stability of the doubly superionic phase of this material. The symbols label the different simulation methods while the colors refer to the four phases.

CH₂N₂-Pna2₁ exhibits a negative offset in pressure with little to no offset in energy upon the hydrogen superionic to doubly superionic transition. This implies that this transition has a negative Clapeyron slope ($dT/dP < 0$) and that the doubly superionic phase is denser. Upon melting, this ice shows little to no offset in pressure but a large offset in energy.

Finally, HCNO-Pca2₁-II shows pressure discontinuities across both phase transitions but exhibits a significant energy offset only upon melting. This ice also shows the highest melting temperature among the four materials, which exceeds conditions for the interiors of Uranus and Neptune that Redmer et al.[6] estimated assuming their mantles were composed of pure H₂O.

All four doubly superionic materials melt at temperatures ~3000 K lower than H₂O, which melts between 8000 and 9000 K in the 400–600 GPa range. The vicinity of the melting conditions of the four doubly superionic materials to the planetary adiabats of Uranus and Neptune imply that novel layerings may exist in the interiors of ice giants. For example, there may exist a sharp boundary between a doubly superionic layer below and a liquid layer above. Conversely, a doubly superionic layer of CH₂N₂-Pna2₁ would float on top of a denser liquid layer of the same composition. Such a boundary would also hinder the planet's convective cooling across this layer, as this material's negative Clayperon slope implies that a colder downwelling slab would penetrate into the liquid layer, where it would be surrounded by denser fluid which would exert a buoyancy force against convection, as[58,59] discuss for the Earth's mantle. The doubly superionic to liquid transitions of the three other ices all have positive Clapeyron slopes and thus would not hinder convection.

## Discussion

In our study of high-pressure H-C-N-O ices[43,44], we have observed double superionicity in four out of thirteen ices: H₃NO₄-P2₁2₁2₁, H₃NO₄-C2₁, CH₂N₂-Pna2₁, and HCNO-Pca2₁-II. We expect that a broader investigation of planetary ices at extreme pressures and temperatures would reveal additional materials with this unique diffusion behavior.

Recent experimental works have focused on the identification and characterization of superionic phases of H₂O ices[26,33,36,37,39]. We suggest that experiments be carried out to attempt to synthesize the H-C-N-O ices which we found to be doubly superionic, and then measure their electrical conductivity at high pressures and temperatures. The doubly superionic phase could be verified by an increase of the electrical

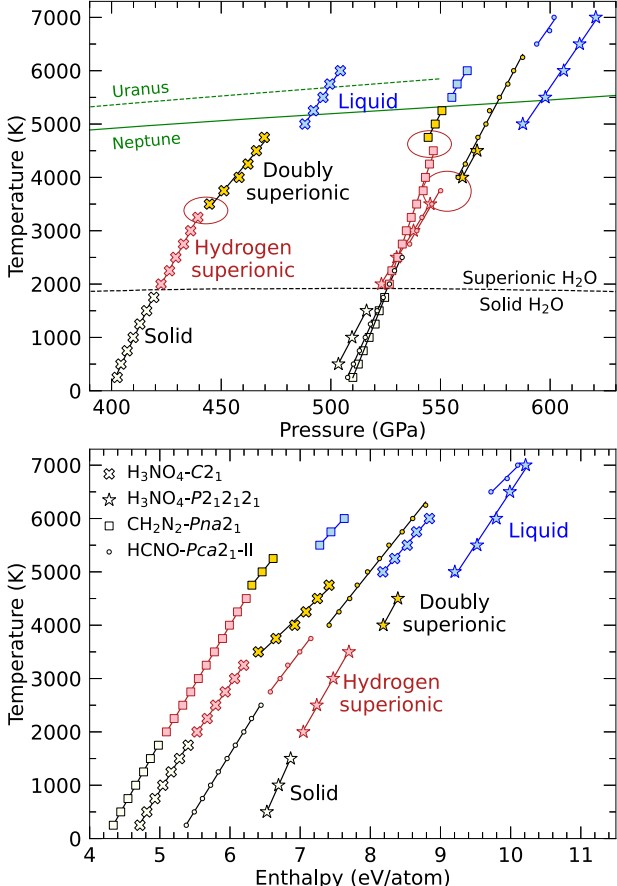

**Fig. 5 | Pressure-temperature and internal energy-temperature trends from simulations at constant density of the four materials that exhibited a doubly superionic state.** The ellipse marks the offsets in pressure between the hydrogen and doubly superionic phases, which implies this transition to be of first order and the phases to be different densities when compared for the same pressure. Planetary isentropes of Uranus and Neptune are from Redmer et al.[6] and the solid to hydrogen superionic transition line for fcc water is from Wilson et al.[27]. The symbols label the four materials while the colors refer to the four phases.

conductivity. The transition from the hydrogen superionic phase to the doubly superionic phase may also be detected using X-ray diffraction, as the peaks associated with the doubly superionic element change from sharp to diffuse[60].

An alternative computational approach which could advance our understanding of the superionic behaviors of planetary ices is path integral molecular dynamics (PIMD) simulations, where nuclei are treated quantum mechanically rather than classically, as done in traditional DFT-MD simulations. Given their low mass, nuclear quantum effects are expected to be strongest for H nuclei, which may lower the transition temperature to a superionic state and potentially heighten the proton diffusivity, as was observed in ref. 61.

Numerical dynamo models[8,9] have shown that the unusual, non-dipolar magnetic fields of Uranus and Neptune observed by Voyager II are most likely produced by convection of charged materials in a thin shell geometry in their outer mantles. Unlike Jupiter and Saturn, Uranus and Neptune do not reach high enough temperatures and pressures for hydrogen to become metallic, so one assumes that their magnetic fields originate from convection of conducting planetary ices. The high ionic conductivity of hydrogen in the superionic phase of water has been shown experimentally to make a substantial contribution to the total electrical conductivity[33]. Assuming a mixture of H₂O, NH₃, and CH₄, Conway et al.[43] and Naumova et al.[44] analyzed what stable compounds form under high pressure conditions in planetary

interiors. Here we show that these materials also demonstrate super-ionic and doubly superionic phases which enhance their electrical conductivity and will thus affect the generation of the planetary dynamos of Uranus and Neptune.

Our results have relevance beyond the ice giant planets in our solar system. Within the last 10 years, technological advances have enabled the indirect observation of exoplanet magnetic fields, including for a Neptune-sized planet[62–64]. We expect an increasing number of such observations to be made for Neptune-sized planets, given the current size distribution of exoplanets discovered[1]. Spectroscopic observations have shown that planet-hosting stars have varying elemental compositions. As planets form within the same accretion disk as their host star and thus inherit a signature of the composition of their star, other solar systems must have ice giant planets with abundances of H, C, N, and O different from Uranus and Neptune. In such planets, superionic phases of planetary ices such as those we have studied would likely influence dynamo generation.

Following recommendations of the most recent Planetary Science Decadal Survey[65], a Uranus Orbiter and Probe mission is now likely. Similar to the Cassini and Juno missions which sent spacecraft to Saturn and Jupiter, respectively, one of the main goals of this upcoming mission to Uranus will be to collect gravitational and magnetic field data of this planet. This data, in combination with equations of state derived from careful study of planetary materials like the ones we present here, will be used to develop more accurate interior profiles for Uranus to enhance our understanding of this planet. In preparation for this mission, it is of critical importance to strengthen our understanding of planetary ices at the extreme conditions that exist in the interiors of ice giant planets.

## Methods
### DFT-MD
DFT calculations were implemented with the Vienna ab initio Simulation Package (VASP)[66]. Electronic states were populated according to a Fermi-Dirac distribution by equating the electronic temperature to that of the ions using the Mermin functional[67], to incorporate the excited electronic states within the Kohn-Sham formalism[68]. We employed projector augmented wave (PAW) pseudopotentials[69] to represent electronic wavefunctions, using a plane wave energy cut-off of 1100 eV and core radii of 0.8, 1.5, 1.5, and 1.1 Å for hydrogen, carbon, nitrogen, and oxygen, respectively; harder pseudopotentials were tested for $H_3NO_4$-$P2_12_12_1$ (Fig. S3) with consistent results.

The Perdew, Burke, and Ernzerhof (PBE) generalized gradient approximation (GGA) was used to model the exchange-correlation energy[70]. The Brillouin zone of the supercells generated were sampled using the Γ-point only, having ensured convergence of internal energy, pressure, and normal stresses to within 1% by performing tests using up to a $4 \times 4 \times 4$ Monkhorst-Pack k-points grid. K-points convergence tests and a sample DFT-MD simulation for $H_3NO_4$-$P2_12_12_1$ using a 2x2x2 k-points grid can be found in Tables S1–4 and Fig. S1.

NVT simulations were performed using a Nosé-Hoover thermostat[71,72]. with calculations for each material completed for a single density and for temperatures ranging from 500 K up to the melting point (2500 K-6500 K), in 500 K increments. Melting points were determined using the heat-until-it-melts method, and thus provide just an upper bound to the possible melting temperature[73]. We used a time step of 0.5 fs in all DFT-MD simulations with durations between 1 ps and 6 ps for simulations at or below 1500 K, and durations between 3 ps and 70 ps at or above 2000 K. Each doubly superionic ice was simulated for a duration of 40–70 ps for at least one temperature exhibiting double superionicity. We performed test simulations with a time step of 0.2 fs and found the internal energies to be consistent with those predicted using a 0.5 fs timestep. Initial simulation cells for the doubly superionic materials contained 96 atoms for $H_3NO_4$-$P2_12_12_1$, 144 atoms for $H_3NO_4$-$C2_1$, 128 atoms for HCNO-$Pca2_1$-II, and 160 atoms for

$CH_2N_2$-$Pna2_1$, with periodic boundary conditions applied. We performed additional simulations for HCNO-$Pca2_1$-II and $H_3NO_4$-$C2_1$ with 304 atom cells, and found that these larger cells well replicated the phase transition temperatures and atomic diffusion behavior observed in the smaller cells (Fig. S2).

Additional simulations were performed on $H_3NO_4$-$P2_12_12_1$ using the NVT, NPT and NPH ensembles using a 288 atom cell with pressures ranging from 450–600 GPa. The nearly-cubic 288 atom cell was constructed using ref. 74. NPT and NPH simulations implemented a Langevin thermostat[75] using a lattice mass of 1000 amu. NPT simulations used friction coefficients for the lattice and atomic degrees of freedom of 10 ps$^{-1}$. All friction coefficients were set to 0 for NPH simulations. NPH simulations were initialized from well-equilibrated structures from NPT simulations of the same temperature and pressure.

Two-phase NVT simulations were performed for $H_3NO_4$-$P2_12_12_1$ using a 192 atom cell for 4000 K, 4500 K, and 5000 K. In this process, a doubly superionic cell and a liquid cell are joined along two interfaces. For 1000 ionic steps only the atoms from the liquid cell are allowed to move, then for 1000 ionic steps only the atoms from the doubly superionic cell are allowed to move. This process allows for the equilibration of the interfaces between the solid and the liquid phases. Finally, all atoms are allowed to move simultaneously and the final state of the cell indicates the more energetically stable phase at that temperature and pressure.

### Machine learning
Machine learning potentials for each doubly superionic compound were constructed using the Deep Potential for Molecular Dynamics (DeePMD) method[55,56], in which a deep neural network is trained using atomic coordinates, potential energies, and forces from many DFT-MD generated configurations, from which it learns the potential energy surface of a chemical system. Such techniques allow for the generation of interatomic potentials with DFT accuracy, but with a computational cost more akin to that of classical potentials. Training and simulation details can be found in Text S3A. See Figs. S12–18 for benchmarks of the machine learning potentials, including training errors, radial distribution functions, and energy/force parities between DFT and ML.

## Data availability

The data generated in this study have been deposited at the Zenodo database under accession code https://doi.org/10.5281/zenodo.10108694. To keep the data volume reasonable, only the files from the first checkpoint of DFT-MD and ML-MD simulations are provided. Further simulation data are available from the corresponding author upon reasonable request.

## Code availability

DFT-MD simulations were performed using The Vienna Ab initio Simulation Package (VASP)[66]. Machine learning potentials were trained with the DeePMD-kit[55,56], and were implemented using the LAMMPS code[57].

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

## Acknowledgements

We thank R. Domingos for advice on generating machine learning potentials from DFT-MD data. We thank V. Naden Robinson for discussions on planetary ices. K.D.V., F.G.-C., and B.M. acknowledge support from the Center for Matter at Atomic Pressures (CMAP) which is funded by the U.S. National Science Foundation (PHY-2020249). Computational resources at the National Energy Research Scientific Computing Center and the Livermore Computing Center were used.

## Author contributions

B.M. conceived of the project; K.D.V. performed all simulations; K.D.V., F.G.-C. and B.M. analyzed the data and wrote the manuscript.

## Competing interests

The authors declare no competing interests.
