## [Peer Review File · Nature Communications]

Double Superionicity in Icy Compounds at Planetary Interior ConditionsREVIEWER COMMENTS

Reviewer #1 (Remarks to the Author):

This paper investigated the dynamic properties of C-N-O-H system at high pressure and high temperature, corresponding to the internal condition of Uranus and Neptune. The authors used ab initio molecular dynamics, combining with machine learning potentials to explore the phase diagram of several C-N-O-H compounds. They claimed that four compounds exhibit double-superionic behavior. The results are clear and the paper is well-written. However, the authors just run MD simulations with the crystal structures found in previous PNAS paper. Therefore, as a follow-up paper, I think that the degree of innovation of this work is not enough for Nature Communications, but rather a more specialized journal. In addition, their analysis on the phenomena of the superionic phases is not detailed enough. To illustrate this point further, I have the following comments:

1. The authors should provide more information on the ML potentials such as the training error data. The accuracy of the ML potentials is essential to the results of this study. Only the comparison of the radial distribution functions for the DFT and ML potentials is not enough.
2. The second sub-figure in Fig. 4. use internal energy as x-axis. However, these compounds are formed by different chemical species (H-N-O, C-H-N, H-C-N-O). Is it suitable to put all of them in the same figure? Also, this sub-figure provides little useful information.
3. The authors claimed that the C-H-O-N compounds may dominate the interior structure of the Uranus and Neptune. What is the abundance of the elements C and N in the interior of ice-giants? The author should provide some related references.
4. Why the critical transition temperatures of the solid phase and superionic phase decrease with proton fraction? The authors need to explain this phenomenon.
5. Can the authors explain why some C-N-O-H compounds exhibit double superionic behavior, while others not?

6. One of the reasons why superionic behavior in the interior of the ice-giants is important is that the superionic behavior of the water may generate the planetary magnetic field. Is it the same for the superionic behavior of the C-H-O-N compounds? It will be valuable if the authors could provide some data reflecting the effect of the C-H-O-N compounds on the generation of the planetary magnetic field.

Reviewer #2 (Remarks to the Author):

The manuscript details ab-initio molecular dynamics simulations on thirteen compounds containing H-C-N-O elements at pressure and temperature conditions relevant to giant icy planets. The authors find that four of the thirteen ices studied exhibited so-called double superionicity, as named by the authors. Here, with increasing temperature, the first superionic state corresponds to H ions diffusing through a lattice of the heavier elements. As temperature is raised further a second species (either C or N) becomes diffusive as well while the remaining heavier atomic species maintain the stable lattice. While this phenomena is hardly surprising, I do not believe it has been discussed or noted in previous literature. Furthermore, machine learning potentials are developed so that longer-scale MD simulations may be performed to study the superionic phases.

The manuscript is really nicely written, with appropriate reference to the literature, discussion of planetary relevance, and compelling graphics. My only suggestion would be to move some of the more technical details regarding the ML training and validation to the SI and save the most important computational details for the main manuscript. Similarly, Eq 1 could also be put into the SI or computational details.

In summary, the manuscript is nicely presented and the work performed well. While I am not surprised by the phenomena discussed, I do not think it has been noted in previous literature.

Reviewer #3 (Remarks to the Author):

This paper describes the development of a machine learned interatomic potential (MLP), trained to reproduce Density Functional Theory energies and forces and used to study the properties of several CHON mixtures at temperatures and pressures relevant to large (Neptune and sub-Neptune) icy planets. The authors report several systems exhibiting super-ionic diffusion but more importantly some systems with not only hydrogen diffusion but also other elements diffusing through a lattice composed of the heavier nuclei. The latter, if true, is quite an interesting find and worthy of publication in Nature Communications. However, this conclusion is based on a set of molecular dynamics simulations performed on very small system sizes in the NVT ensemble, and it is quite possible under those constraints (small system size, NVT) to over-stabilize the solid phase.

The main advantage of developing MLPs is to overcome both time and length-scales limitations of DFT-MD. The authors use their MLPs to explore longer simulation times but not larger system sizes. To convincingly show that the doubly super-ionic phases are real the authors need to show that these phases are 1) more energetically stable than the liquid phase and 2) also stable against (supercell) deformation. There is also 3) the issue of the chemical composition of the liquid versus the solid which may not be the same.

To answer condition 1, the authors could perform coexistence simulations in the NPH ensemble with a large number of atoms to establish the melting phase lines with more accuracy (or they could use the interface pinning method or any other appropriate free energy comparison technique). The heat-until-melt method used in this study is not state-of-the-art given the MLP that was developed.

To answer condition 2, single phase NPT or NPH simulations might suffice but even more information could be gathered from meta-dynamics simulations to ascertain free energy differences between various solid phases. These capabilities are included in the LAMMPS code the authors used in their study.

While issue 3 is important and key to planetary applications, this could be part of further studies as it would represent a much larger scope of work.

On a more technical aspect, it would be useful to show more details about the fitted MLPs in the supplementary information section, e.g. force parity plots (MLP vs DFT), k-point convergence of the forces, comparison of the average and variance of the energy between DFT and the MLP, comparisons of vibrational density of states between the MLP and DFT.

> **Reviewer #1 (Remarks to the Author):**

> This paper investigated the dynamic properties of C-N-O-H system at
> high pressure and high temperature, corresponding to the internal
> condition of Uranus and Neptune. The authors used ab initio molecular
> dynamics, combining with machine learning potentials to explore the
> phase diagram of several C-N-O-H compounds. They claimed that four
> compounds exhibit double-superionic behavior. The results are clear
> and the paper is well-written. However, the authors just run MD
> simulations with the crystal structures found in previous PNAS
> paper. Therefore, as a follow-up paper, I think that the degree of
> innovation of this work is not enough for Nature Communications, but
> rather a more specialized journal.

We thank the reviewer for their thoughtful review of our manuscript and their suggestions.

We believe that this work, describing the first prediction with simulations of double superionicity in H-C-N-O ices, is a significant advancement towards the understanding of planetary ices and is thus ideal for publication in Nature Communications. Through our unprecedentedly wide investigation of the diffusive properties of H-C-N-O ices, we have repeatedly observed a new state of matter, which we believe to exist in ice giant interiors, and which could potentially influence layering and convective patterns of these planets. In our opinion, this renders our results important, a view which is supported by the positive feedback of Reviewers 2 and 3.

> In addition, their analysis on the
> phenomena of the superionic phases is not detailed enough. To
> illustrate this point further, I have the following comments:

> 1. The authors should provide more information on the ML potentials
> such as the training error data. The accuracy of the ML potentials is
> essential to the results of this study. Only the comparison of the
> radial distribution functions for the DFT and ML potentials is not
> enough.

We agree that proper testing of our machine learning potentials is imperative to the results of our study. In our updated supplemental file we have added data from a number of relevant tests, including: a detailed description of our machine learning training process which we hope will help future researchers succeed in training their own potentials, RMSE plots of energy and force training and validation errors for H₃NO₄-P212121 (**Fig. S12**), force and energy parity plots between DFT and DeePMD for H₃NO₄-P212121 (**Fig. S13**), comparisons of the average energies and energy variances between DFT and DeePMD for H₃NO₄-P212121 (**Fig. S14**), and finally, radial distribution function comparisons between DFT and DeePMD simulations for H₃NO₄-P212121 for the solid, superionic, doubly superionic, and liquid phases (**Figs. S15-18**).

> 2. The second sub-figure in Fig. 4. use internal energy as
> x-axis. However, these compounds are formed by different chemical
> species (H-N-O, C-H-N, H-C-N-O). Is it suitable to put all of them in
> the same figure? Also, this sub-figure provides little useful
> information.

We agree with the reviewer in the sense that comparing the absolute internal energies between different materials may not be the most physically meaningful quantity, as the energy of each compound is relative to the ground state energies of its constituent elements, as in VASP convention. We did not plot the compounds together to emphasize the relative energies of each material; but rather to emphasize relative energies of different phases of a given material, and how energies of individual phases change with temperature.

We believe that using internal energy vs. temperature can give us an idea of how the specific heat of each material changes upon each phase transition, because this quantity corresponds to the slope of such isochores. However, we have replaced the energy-temperature figure with an enthalpy-temperature figure. In this plot, we can observe the enthalpy input required for each phase change to occur. While the simulations shown were performed in the NVT ensemble, the observed enthalpy difference between phases is analogous to the latent heat between phases in the NPT ensemble, and serves to further differentiate the doubly superionic phase from both the hydrogen superionic and the liquid phases.

We feel that both the pressure-temperature and enthalpy-temperature plots are required to adequately distinguish all phases of a given material. For example, *CH₂N₂-Pna21* shows a discontinuity in P-T space but not in H-T space between the hydrogen superionic and doubly superionic state. However, between the doubly superionic and liquid states, this material has no P-T discontinuity but does have a H-T discontinuity .

We have copied the updated Figure 5 below.

- > 3. The authors claimed that the C-H-O-N compounds may dominate the interior structure of the Uranus and Neptune. What is the abundance of the elements C and N in the interior of ice-giants? The author should provide some related references.

It has been historically argued that Uranus and Neptune are “ice” rich due to their formation beyond the frost line, nominally allowing them to accrete large amounts of planetary ices as they formed. Assuming solar abundances of oxygen to be present in these planets, the ice to rock ratio would be 2:3. It has been further assumed that planetary ices in these planets overall contain solar abundances of H:O:C:N of ~28:7:4:1 (a mixture called “Synthetic Uranus”).

We have included the above discussion and the following references in the main text:

Helled, R., Fortney, J.J.: The interiors of Uranus and Neptune: current understanding and open questions. *Philosophical Transactions of the Royal Society A* 378(2187), 20190474 (2020).

Radousky, H., Mitchell, A., Nellis, W.: Shock temperature measurements of planetary ices: NH₃, CH₄, and “synthetic uranus”. *The Journal of Chemical Physics* 93(11), 8235–8239 (1990)

Chau, R., Hamel, S., Nellis, W.J.: Chemical processes in the deep interior of Uranus. *Nature Communications* 2(1), 203 (2011)

- > 4. Why the critical transition temperatures of the solid phase and superionic phase decrease with proton fraction? The authors need to explain this phenomenon.

We agree with the reviewer that this phenomenon warranted further investigation. With the reviewer’s request in mind we studied the atomic environments of each material and found that the temperature at which H ions begin to diffuse in each material is correlated to the number of heavy nuclei (C, N, and O) around the H ions. As the proton fraction decreases, the number of heavy elements around H ions at short distances increases, causing the H ions to become increasingly constrained, and as a result H ions must reach higher temperatures before they acquire the kinetic energy required to escape their potential wells. From this analysis, we have added Fig. 1b (copied below) to our manuscript.

- > 5. Can the authors explain why some C-N-O-H compounds exhibit double superionic behavior, while others not?

At the present time, we are not able to offer an explanation as to why some compounds exhibit doubly superionic behavior while others do not. One has to perform DFT-MD simulations to determine the superionic behavior of these materials. The goal of our manuscript is to show that double superionicity exists in H-C-N-O compounds and it is a stable, well-defined state. We

would like to point out that there is no accepted theory to predict whether compounds go singly-superionic before they melt. Again one has to perform DFT-MD simulations for every single material. Unfortunately, we are not able to provide something that the entire community of superionic research has not been able to achieve so far, but we strongly believe that further studies are required and we hope to motivate such studies.

- > 6. One of the reasons why superionic behavior in the interior of the
- > ice-giants is important is that the superionic behavior of the water
- > may generate the planetary magnetic field. Is it the same for the
- > superionic behavior of the C-H-O-N compounds? It will be valuable if
- > the authors could provide some data reflecting the effect of the
- > C-H-O-N compounds on the generation of the planetary magnetic field.

We agree with the reviewer that data on the conducting properties of planetary ices, particularly those outside of stoichiometric mixtures of ammonia and water, is sparse. Moreover such data is of critical importance to understanding the possible role of H-C-N-O materials on magnetic field generation in ice giant planets. With this in mind, we calculated the H ion diffusivities, D , for each of the 13 H-C-N-O materials we studied and found a linear trend of $\ln(D)$ vs. $1/T$, indicating the diffusion rates of these materials follow a classic Arrhenius trend, as shown in the newly added Figure 1c (below). Interestingly, we found that materials which became superionic at a higher temperature had a steeper slope, corresponding to a higher activation energy required for H ions to diffuse.

Furthermore, we used these diffusivities to calculate the ionic conductivity of each of the 13 H-C-N-O ices using the Nernst-Einstein relation. H ionic charges were estimated by first calculating the Bader electronic charge on each H ion and then by calculating the resultant net positive charge on each H nucleus (**Fig. S4**). Figure 1d has been added to show these conductivities. We find the conductivities of these planetary ices to be comparable to high pressure water and ammonia. The high ionic conductivities of the superionic phases of these H-C-N-O materials suggests that these or similar H-bearing ices could be important for planetary dynamos if they were present in the proposed convecting region of ice giant planets responsible for magnetic field generation.

The updated Figure 1 is copied below, and a full description of our findings which we detailed above has been added to the main text.

> Reviewer #2 (Remarks to the Author):

>

> The manuscript details ab-initio molecular dynamics simulations on
> thirteen compounds containing H-C-N-O elements at pressure and
> temperature conditions relevant to giant icy planets. The authors find
> that four of the thirteen ices studied exhibited so-called double
> superionicity, as named by the authors. Here, with increasing
> temperature, the first superionic state corresponds to H ions
> diffusing through a lattice of the heavier elements. As temperature is
> raised further a second species (either C or N) becomes diffusive as
> well while the remaining heavier atomic species maintain the stable
> lattice. While this phenomena is hardly surprising, I do not believe
> it has been discussed or noted in previous literature. Furthermore,
> machine learning potentials are developed so that longer-scale MD
> simulations may be performed to study the superionic phases.

>

> The manuscript is really nicely written, with appropriate reference to
> the literature, discussion of planetary relevance, and compelling
> graphics. My only suggestion would be to move some of the more
> technical details regarding the ML training and validation to the SI
> and save the most important computational details for the main
> manuscript. Similarly, Eq 1 could also be put into the SI or
> computational details.

We thank reviewer 2 for their positive comments on our manuscript.

Technical details of the ML training and validation have been moved to the SI. Only a brief description of the process remains in the main text.

We prefer to leave Equation 1 in the main text, as many of our analyses are reliant on the concepts of mean squared displacement and diffusion coefficients and we want to ensure all readers' familiarity with the calculation.

> In summary, the manuscript is nicely presented and the work performed
> well. While I am not surprised by the phenomena discussed, I do not
> think it has been noted in previous literature.

> Reviewer #3 (Remarks to the Author):

>

> This paper describes the development of a machine learned interatomic potential (MLP), trained to reproduce Density Functional Theory energies and forces and used to study the properties of several CHON mixtures at temperatures and pressures relevant to large (Neptune and sub-Neptune) icy planets. The authors report several systems exhibiting super-ionic diffusion but more importantly some systems with not only hydrogen diffusion but also other elements diffusing through a lattice composed of the heavier nuclei. The latter, if true, is quite an interesting find and worthy of publication in Nature Communications.

>

> However, this conclusion is based on a set of molecular dynamics simulations performed on very small system sizes in the NVT ensemble, and it is quite possible under those constraints (small system size, NVT) to over-stabilize the solid phase.

>

> The main advantage of developing MLPs is to overcome both time and length-scales limitations of DFT-MD. The authors use their MLPs to explore longer simulation times but not larger system sizes. To convincingly show that the doubly super-ionic phases are real the authors need to show that these phases are 1) more energetically stable than the liquid phase and 2) also stable against (supercell) deformation. There is also 3) the issue of the chemical composition of the liquid versus the solid which may not be the same.

We thank the reviewer for their insightful and thorough comments on our manuscript.

With the reviewer's concerns and suggestions in mind, we have performed a number of additional simulations, using both DFT and ML methods, to demonstrate the energetic and structural stability of the doubly superionic phase.

> To answer condition 1, the authors could perform coexistence simulations in the NPH ensemble with a large number of atoms to establish the melting phase lines with more accuracy (or they could use the interface pinning method or any other appropriate free energy comparison technique). The heat-until-melt method used in this study is not state-of-the-art given the MLP that was developed.

We agree with the reviewer on the importance of demonstrating the energetic stability of the doubly superionic phase relative to the liquid phase. Prompted by the reviewer, we have performed two-phase simulations for one of the doubly superionic materials (H₃NO₄-P212121) and we were able to successfully crystallize a liquid into a doubly superionic state. In this

process, a liquid cell and doubly superionic cell were combined into one supercell with two liquid-doubly superionic interfaces. Initially, the atoms of the liquid were briefly allowed to move while the atoms of the doubly superionic phase were held still, followed by the atoms of the doubly superionic phase being allowed to move while the liquid atoms were held still, to allow for equilibration of the liquid-doubly superionic interfaces. Thereafter all atoms were allowed to move. For the two temperatures predicted to be doubly superionic by all other simulation methods (4000 K and 4500 K), the two-phase simulations successfully crystallized the oxygen atoms out of the liquid phase into an fcc lattice. For the lowest temperature predicted to be liquid by the other DFT-MD simulations, the liquid phase melted the doubly superionic phase. Our two-phase simulations well replicated the melting temperature obtained from our other simulation methods. We have included **Figures S10 and S11** for visualizations of this simulation at 4000 K and ~500 GPa, which are copied here below. We have also included **Movie S2** showing the crystallization of the liquid oxygen atoms into an fcc sublattice in this simulation.

- > To answer condition 2, single phase NPT or NPH simulations might
- > suffice but even more information could be gathered from meta-dynamics
- > simulations to ascertain free energy differences between various solid
- > phases. These capabilities are included in the LAMMPS code the authors
- > used in their study.

Prompted by the reviewer, we have also performed numerous additional NVT, NPT, and NPH simulations with 288 atom cells for H₃NO₄-P212121, to confirm that the doubly superionic phase is stable when simulated with a larger system size and when subjected to cell deformation. We found that all simulation methods across a wide range of pressures and temperatures are in excellent agreement with each other, and with our ML and two-phase simulations.

As an example of our successful NPT simulations, we have included **Figures S8** and **S9** below, which show the MSD and thermodynamic data from a simulation of H₃NO₄-P212121 at 525 GPa and 4500 K. Even with cell fluctuations, N and H diffuse consistently through a stable fcc O sublattice.

Additionally, we have been able to successfully replicate the doubly superionic behavior of this material with ML simulations of 1920 atoms and simulation times over 100 ps. We have included a video of one such simulation (see **Movie S1**). Below is a snapshot from this simulation showing the trajectory of N ions through the stable fcc O sublattice. H ions are excluded to allow for easier visualization of the N trajectories:

The numerous pressure-temperature conditions simulated through our NVT, NPT, NPH, ML, and two-phase simulations we have performed for H_3NO_4 - $P212121$ following our receipt of the reviewer's original comments are all plotted below, along with our original trajectory of the 96 atom cell for this material. We see that using the heat-until-it-melts method for the 96 atom trajectory overshoot the melting line, as expected for this approach, but the other simulation methods are all in excellent agreement. The ML potential using 1920 atoms also agrees well with the DFT data, with the only discrepancy being that it predicts that the material should become doubly superionic at 3500 K - this can easily be explained as a slightly unstable phase finally transitioning following the long timescale involved in this simulation. This figure below, excluding the 96 atom trajectory, has been added to the manuscript.

While we are intrigued by the high pressure community's growing use of metadynamics, and we are sure that such a method will prove invaluable to our group in the future as machine learning techniques become more commonplace, we feel for the time being that such methods are outside the scope of this study.

We believe that the numerous congruent results from our NVT, NPT, NPH, and two-phase simulations using DFT, and our large cell ML simulations, unambiguously confirm the stability of the doubly superionic phase.

> While issue 3 is important and key to planetary applications, this
 > could be part of further studies as it would represent a much larger
 > scope of work.

We thank the reviewer for this feedback. We agree, this is an important issue that has relevance for planetary science and it is within our interests to explore this point further. Given the extensive possibilities for further investigation, we believe that delving into the study of the chemical composition of the liquid in greater depth would entail a significantly broader scope of work. We are eager to dedicate our future studies to this endeavor. In addition, we think that our work also has the potential to motivate other scientists to pursue such studies as well. In this

particular work, we focused on reporting the existence of these new phases at high temperatures and provide background for future research.

- > On a more technical aspect, it would be useful to show more details
- > about the fitted MLPs in the supplementary information section,
- > e.g. force parity plots (MLP vs DFT), k-point convergence of the
- > forces, comparison of the average and variance of the energy between
- > DFT and the MLP, comparisons of vibrational density of states between
- > the MLP and DFT.

We agree that any work which draws conclusions from machine learning simulations must be accompanied by extensive testing of these potentials. Please see our updated supplemental file for:

- K-point convergence tests for DFT-MD simulations for all doubly superionic materials (**Tables S1-S4**)
- MSD comparison for H3NO4-*P212121* at 500 GPa, 4500K, using a 1x1x1 and 2x2x2 k-point grid (**Fig. S1**)
- A full description of our machine learning training process
- RMSE plots of energy and force training and validation errors for H3NO4-*P212121* (**Fig. S12**)
- Force and energy parity plots between DFT and DeePMD for H3NO4-*P212121* (**Fig. S13**)
- Comparison of average energies and energy variances between DeePMD and DFT for H3NO4-*P212121* (**Fig. S14**)
- Radial Distribution Function comparisons between DFT and DeePMD for H3NO4-*P212121* for the solid, superionic, doubly superionic, and liquid phases (**Figs. S15-S18**)

REVIEWER COMMENTS

Reviewer #1 (Remarks to the Author):

I appreciate the opportunity to review the revised manuscript, and I commend the authors for their efforts in addressing our previous concerns and making appropriate modifications to their work. However, after careful consideration, I still have reservations about the suitability of this manuscript for publication in Nature Communications. My concerns are as follows:

Relevance to Planetary Science: Considering the low abundance of nitrogen (N) in ice giants, as indicated in the authors' response, with an H:O:C:N ratio of approximately 28:7:4:1, I have concerns regarding the practical significance of superionic behavior in H-O-C-N compounds for planetary interiors. To address this issue, the authors could discuss the potential implications or applications of their findings within the context of ice giants or other celestial bodies, thereby strengthening the broader scientific impact of their work.

Lack of Depth: The manuscript primarily relies on molecular dynamics (MD) simulations conducted on structures previously reported in a PNAS paper. While these simulations may be suitable for certain research questions, they may be deemed relatively too simple for a journal of Nature Communications' caliber. I recommend that the authors expand on the limitations of their methodology and propose potential avenues for more advanced simulations or analyses that could enhance the depth and innovation of their study.

In light of these concerns, I regret to inform you that I still do not believe that this manuscript meets the rigorous standards expected for publication in Nature Communications, even after the authors' diligent revisions. However, I would like to emphasize that my feedback is intended to help the authors further improve their work. They may consider addressing these points and exploring potential avenues for enhancing the impact of their research before resubmitting it to a suitable journal.

Reviewer #3 (Remarks to the Author):

The authors have adequately addressed my comments and questions. I'm recommending publication of this new version of the manuscript and supporting information.

> Reviewer #1 (Remarks to the Author):

> I appreciate the opportunity to review the revised manuscript, and I commend the authors for
> their efforts in addressing our previous concerns and making appropriate modifications to their
> work.

> However, after careful consideration, I still have reservations about the suitability of this
> manuscript for publication in Nature Communications. My concerns are as follows:

> Relevance to Planetary Science: Considering the low abundance of nitrogen (N) in ice giants,
> as indicated in the authors' response, with an H:O:C:N ratio of approximately 28:7:4:1, I have
> concerns regarding the practical significance of superionic behavior in H-O-C-N compounds
> for planetary interiors. To address this issue, the authors could discuss the potential
> implications or applications of their findings within the context of ice giants or other celestial
> bodies, thereby strengthening the broader scientific impact of their work.

We thank Reviewer #1 for their time and feedback on our manuscript. We have carefully considered the concerns raised and would like to address them, highlighting the importance of our findings on superionic behaviors of H-C-N-O materials for planetary science.

As we have discussed in our paper, and as the reviewer references, we study H-C-N-O materials out of an interest for ice giant interiors, as we expect ice giant planets to be dominated by these elements after accreting large amounts of solid H₂O, NH₃, and CH₄ during their formation beyond the frost line of our Solar System. Although we assume the bulk compositions of Uranus and Neptune to have solar ratios of H:O:C:N of 28:7:4:1, the actual abundances could be different and there are many mechanisms by which chemical separation may occur in these planets. This would result in regions of these planets that are enriched or depleted in various elements relative to solar abundances. We observe a similar phenomenon with the chemical differentiation present throughout the Earth: the relative scarcity of iron in the crust and the mantle belies the Earth's true iron content. In ice giant planets, differentiation processes likely also occur. As just one example, many previous works have suggested the possibility of diamond formation at extreme pressures and temperatures within Uranus and Neptune, which would result in the segregation and sinking of dense carbon and bubbling up of hydrogen, a phenomenon usually referred to as "diamond rain."

While we cannot state with certainty what materials would form under planetary conditions from a mixture of water, methane, and ammonia, previous works have shown that methane becomes unstable under planetary interior conditions, and that although water and ammonia are stable to decomposition at extreme temperatures and pressures, the high pressure ammonia-hydrate compounds they form are more energetically favorable. We must then assume that the interiors of ice giant planets are dominated by the high pressure transformation products of these compounds. The pioneering works of Conway *et al.* (2021) and Naumova *et al.* (2021), which reported the first structure searches of the full H-C-N-O quaternary space, give us the most plausible predictions so far of what materials may actually exist in these planets. Our work reports an extensive and thorough investigation of these materials at the temperatures relevant

to ice giant interiors and provides the most plausible assumptions in the current literature for how planetary ices behave in Uranus, Neptune, and sub-Neptune exoplanets.

Previous works (Stanley & Bloxham; 2004, 2006) have shown that the unusual, non-dipolar magnetic fields of Uranus and Neptune observed by Voyager II are most likely produced by convection of charged materials in a thin shell geometry in their outer mantles. Unlike Jupiter and Saturn, Uranus and Neptune do not reach high enough temperatures and pressures for hydrogen to become metallic, so one assumes that their magnetic fields originate from convection of conducting planetary ices. The high ionic conductivity on hydrogen in the superionic phase of water has been shown experimentally to make a substantial contribution to the total electrical conductivity (Millot *et al.* 2018). Assuming a mixture of water, ammonia, and methane, Conway *et al.* and Naumova *et al.* analyzed what stable compounds form under high pressure conditions in planetary interiors. Here we show that these materials also demonstrate superionic and doubly superionic phases which enhance their electrical conductivity and will thus affect the generation of the planetary dynamos of Uranus and Neptune.

Our results have relevance beyond the ice giant planets in our solar system. Within the last 10 years, technological advances have enabled the indirect observation of exoplanet magnetic fields, including one for a Neptune-sized planet. We expect an increasing number of such observations to be made for Neptune-sized planets, given the current size distribution of exoplanets discovered. Spectroscopic observations have shown that planet-hosting stars have varying elemental compositions. As planets form within the same accretion disk as their host star and thus inherit a signature of the composition of their star, other solar systems must have ice giant planets with abundances of H, C, N, and O different from Uranus and Neptune. In such planets, superionic phases of planetary ices such as those we have studied would likely influence dynamo generation.

We have added the above discussion (highlighted in yellow) and the following references to our manuscript:

Ben-Jaffel, L., Ballester, G. E., Muñoz, A. G., Lavvas, P., Sing, D. K., Sanz-Forcada, J., ... & López-Morales, M. (2022). Signatures of strong magnetization and a metal-poor atmosphere for a Neptune-sized exoplanet. *Nature Astronomy*, 6(1), 141-153.

Kislyakova, K. G., Holmström, M., Lammer, H., Odert, P., & Khodachenko, M. L. (2014). Magnetic moment and plasma environment of HD 209458b as determined from Ly α observations. *Science*, 346(6212), 981-984.

Cauley, P. W., Shkolnik, E. L., Llama, J., & Lanza, A. F. (2019). Magnetic field strengths of hot Jupiters from signals of star–planet interactions. *Nature Astronomy*, 3(12), 1128-1134.

> Lack of Depth: The manuscript primarily relies on molecular dynamics (MD) simulations
> conducted on structures previously reported in a PNAS paper. While these simulations may
> be suitable for certain research questions, they may be deemed relatively too simple for a
> journal of Nature Communications' caliber. I recommend that the authors expand on the
> limitations of their methodology and propose potential avenues for more advanced simulations
> or analyses that could enhance the depth and innovation of their study.

The reviewer is correct in that our analyses rely on MD simulations that were performed on structures previously reported to be stable by the recent works of Conway *et al.* in *PNAS* and Naumova *et al.* in *J. Phys. Chem. A*. However, through these analyses we have repeatedly discovered a new state of matter, double superionicity, within planetary ices, which we believe will hold strong interest in both the planetary science and materials science communities, a view that was shared by Reviewers #2 and #3.

In addition, we have seen that it is common practice for novel analyses to be performed on previously reported structures, even for journals of the caliber of Nature Communications. To name just a few in this journal and sister journal Nature Physics from recent years:

1. Reinhardt, A., Bethkenhagen, M., Coppari, F., Millot, M., Hamel, S., & Cheng, B. (2022). Thermodynamics of high-pressure ice phases explored with atomistic simulations. *Nature Communications*, 13(1), 4707.
2. Cheng, B., Bethkenhagen, M., Pickard, C. J., & Hamel, S. (2021). Phase behaviours of superionic water at planetary conditions. *Nature Physics*, 17(11), 1228-1232.
3. Stixrude, L., Scipioni, R., & Desjarlais, M. P. (2020). A silicate dynamo in the early Earth. *Nature Communications*, 11(1), 935.
4. Belonoshko, A. B., Fu, J., Bryk, T., Simak, S. I., & Mattesini, M. (2019). Low viscosity of the Earth's inner core. *Nature Communications*, 10(1), 2483.
5. Soubiran, F., & Militzer, B. (2018). Electrical conductivity and magnetic dynamos in magma oceans of Super-Earths. *Nature Communications*, 9(1), 3883.
6. Sun, J., Clark, B. K., Torquato, S., & Car, R. (2015). The phase diagram of high-pressure superionic ice. *Nature Communications*, 6(1), 8156.

We do not agree that performing simulations on previously reported structures indicates a lack of depth of our work, but rather that by building off of previous works we are engaging in and furthering the community dialogue on properties of planetary ices. Indeed, both Naumova and Conway in their papers indicated that the study of these materials at high temperature was an important next step to understanding the dynamic properties of planetary ices.

The reviewer is also correct in that our evidence for the existence of the doubly superionic state is based on *ab initio* and machine learning molecular dynamics simulations. With the goal of exploring and verifying various diffusive properties within H-C-N-O materials, we think that molecular dynamics simulations were the most appropriate technique. We do not believe that our focus on this simulation method indicates a lack of rigor in our work; rather, to ensure the veracity of our results we have inspected these materials using an extraordinarily thorough

variety of MD techniques, not present in the aforementioned publications that, in addition, mostly focus on just one material. Through these simulations we addressed satisfactorily all suggestions made by Reviewer #3 following the original submission of our manuscript, to ensure the structural and energetic stability of the doubly superionic phase.

To summarize the simulations we have performed:

1. DFT-MD using NVT ensemble for 13 unique materials using ~100-150 atom cells
2. DFT-MD using NVT ensemble for HCNO-Pca21-II and H3NO4-C21 using ~300 atom cells
3. DFT-MD using NVT, NPT, NPH ensembles for H3NO4-P212121 for 4-5 pressures for 288 atom cells
4. DFT-MD two-phase simulations for H3NO4-P212121
5. ML training and MD simulations for all 4 doubly superionic materials for long timescales
6. ML training and MD simulations for H3NO4-P212121 for large system sizes (1000s of atoms) and long time scales (100s of ps)

We hope that our work will prompt additional investigations of high pressure planetary ices, and we have added suggestions for a few avenues for future research to our manuscript (highlighted in yellow), as suggested by the referee.

1. Related to our prediction of double superionicity in several H-C-N-O ices, we would suggest that experiments be carried out to attempt to synthesize these materials and then measure their electrical conductivity under high pressures and temperatures. Thus, we would have an experimental verification of the existence of the doubly superionic phase. We predict that a discontinuity of the electrical conductivity can be attributed to this phase transition. The doubly superionic phase may also be identified using X-ray diffraction, with the presence of a stable sublattice detected along with a diffuse peak of the superionically diffusing C or N. [See Ref: Kraus 2021 - Melting of Tantalum at Multimegabar Pressures on the Nanosecond Timescale]
2. An alternative computational approach which could advance our understanding of the superionic behaviors of planetary ices is path integral molecular dynamics (PIMD) simulations, where nuclei are treated quantum mechanically as opposed to assuming they behave classically, as in traditional DFT-MD simulations. H nuclei are known to behave more quantum mechanically than other nuclei given their comparatively low mass. Thus we expect that using such methods one would observe heightened proton diffusivity compared to DFT-MD at a given temperature and density; and possibly proton diffusivity at even lower temperatures, as was observed in Ref. [Wang 2021 - Quantum and Classical Proton Diffusion in Superconducting Clathrate Hydrides].

> In light of these concerns, I regret to inform you that I still do not believe that this manuscript
> meets the rigorous standards expected for publication in Nature Communications, even after
> the authors' diligent revisions. However, I would like to emphasize that my feedback is
> intended to help the authors further improve their work. They may consider addressing these
> points and exploring potential avenues for enhancing the impact of their research before
> resubmitting it to a suitable journal.

We appreciate the efforts of the reviewer and their repeated suggestions for strengthening our manuscript. However, we respectfully disagree with this assessment.

Nature Communications is a venue to convey original ideas that have the potential to significantly impact the scientific community, a criteria which we believe we satisfy. By introducing doubly superionic materials in planetary ices, we are combining condensed matter physics with planetary science, an interdisciplinary approach highly valued by Nature Communications.

We present our findings with clearly explained methodologies that can be replicated by other scientists, and we have rigorously confirmed our findings: the doubly superionic phase has been observed in multiple materials and has had its existence and stability rigorously confirmed using many simulation methods, including all of those suggested by the reviewers. Moreover, we have completed an unprecedentedly thorough investigation of the hydrogen superionic behavior of a wide variety of planetary ices. Together, the hydrogen and doubly superionic phases may be important for stratification, convection patterns, and magnetic field generation within ice giant planets, such as Uranus, Neptune, and Neptune-like exoplanets.

Following recommendations presented in the most recent Planetary Science Decadal Survey, it is likely that the next NASA Flagship Mission will be a Uranus Orbiter and Probe. Similar to the Cassini and Juno missions which sent spacecraft to Saturn and Jupiter, respectively, one of the main goals of this upcoming mission to Uranus will be to collect gravitational and magnetic field data of this planet. This data, in combination with equations of state derived from careful study of planetary materials, like the ones we present here, will be used to develop more accurate interior profiles for Uranus to enhance our understanding of this planet. As ice giant planets are expected to be dominated by H-C-N-O ices, it is of critical importance in the coming decades, while we await this spacecraft data, to strengthen our understanding of the material properties of planetary ices and their behaviors at extreme temperatures and pressures. We believe that this work is a significant contribution to that endeavor.

We feel that our work embodies the impactful science that Nature Communications seeks to publish, particularly in light of the opportune timing of our work in relation to growing interest in Uranus and Neptune, the novelty of our discovery of the doubly superionic phase, and the cross-disciplinary importance of our findings to the planetary science and materials science communities.

We kindly request the referee's reconsideration of our manuscript in light of these clarifications.

> Reviewer #3 (Remarks to the Author):

- > The authors have adequately addressed my comments and questions. I'm recommending
- > publication of this new version of the manuscript and supporting information.

We thank the reviewer again for their time and their thoughtful comments on our manuscript.